

# Urban source term estimation for mercury using a boundary-layer budget method

Basil Denzler[1], Christian Bogdal[1], Cyrill Kern[1], Anna Tobler[1,2], Jing Huo[1], and Konrad Hungerbühler[1]

[1]Institute for Chemical and Bioengineering, ETH Zurich, Vladimir-Prelog-Weg 1, CH-8093 Zurich, Switzerland
[2]present address: Paul Scherrer Institute, CH-5232 Villigen PSI, Switzerland

**Correspondence:** Christian Bogdal (christian.bogdal@chem.ethz.ch)

**Abstract.** Mercury is a heavy metal of particular concern due to its adverse effects on the human health and the environment. Recognizing this problem, the UN Minamata Convention on Mercury was recently adopted, where signatory countries agreed to reduce anthropogenic mercury emissions. To evaluate the effectiveness of the convention, quantitative knowledge on mercury emissions is crucial. So far, bottom-up approaches have successfully been applied to quantify mercury emission – especially

for point sources. Distributed sources make up for a large share of the emission, however, they are still poorly characterized. Here, we present a top-down approach to estimate mercury emissions based on atmospheric measurements in the city of Zurich, Switzerland. While monitoring the atmospheric mercury concentrations during inversion periods in Zurich, we were able to relate the concentration increase to the mercury emission strength of the city using a box model. By the means of this boundary-layer budget approach, we succeeded to narrow down the emissions of Zurich to range between $41\pm8$ kg/a (upper bound) and

$24\pm8$ kg/a (lower bound). Thereby, we could quantify emissions from mixed, diffuse and point like sources and derive an annual mercury per capita emission of 0.06 to 0.10 g/a. The approach presented here has the potential to support authorities in setting up inventories and to validate emission estimations derived from the commonly applied bottom-up approaches. Furthermore, our method is applicable to other compounds and to a wide range of cities or other areas, where sources or as well sinks for mercury and other atmospheric pollutants are presumed.

*Copyright statement.* TEXT

## 1   Introduction

The UN Minamata Convention on Mercury entered into force in August 2017. It marks a milestone in the ambitions of the global community to protect the human health and the environment from the adverse effects of mercury and mercury compounds. The parties to this convention have agreed to control mercury emissions and establish an inventory of emissions

from relevant sources. Furthermore, the convention recognizes the need for research and monitoring to increase the state of knowledge regarding the emission and distribution pathways of mercury. Numerous measurement campaigns for atmospheric mercury have been conducted worldwide. The tools applied in the analysis of these measurement results encompass among





others: wind rose interpretations, back trajectories and various statistical analysis. Potential sources for mercury can thereby be located but a quantification of their emission strength is not achieved. For testing the effectiveness of an international treaty such as the Minamata convention, however, quantitative information on emission and changes thereof are crucial. This task of assessing the sources is often left to authorities which follow the guidelines for the bottom-up approach AMAP/UNEP (2013).

The inventories established with this method are very valuable and so far certainly provide the best and most reliable emission estimates. However, these inventories address emission usually on a national level and focus heavily on large point sources. Additionally, the apportionment of total national emissions to regional emissions is a difficult task, requiring various assumptions.To obtain spatially resolved emissions two steps are applied. First, mercury emissions are assigned to point sources where possible. And second, so called 'distributed sources', that make up for >80% of total emissions worldwide, are mapped using

a surrogate on the basis of population density data (Wilson et al., 2006; AMAP/UNEP, 2013). Top-down studies confirming the allocation praxis of distributed mercury emissions in bottom-up inventories are lacking. For scientific requirements, verification and testing of these inventories with other independent methods is necessary, as has similarly been suggested for greenhouse gases (Nisbet and Weiss, 2010). In this work, we present a top-down method that allows quantification of mercury emissions in an urban environment. The goal is to support authorities with the interpretation of their valuable and expensive

monitoring studies and and to ultimately introduce a top-down method that could help to verify and refine mercury emission inventories regarding distributed emissions. To achieve this refinement, the method has to be applicable to numerous locations worldwide with limited resources. For our model set-up we make use of the meteorological phenomenon of a ground inversion. The reduced vertical mixing during high-pressure winter periods or summer nights above ground leads to an accumulation of atmospheric pollutants below the boundary air layer. Inversion effects occur frequently in metropolitan areas all over the world,

such as: Los Angeles, Beijing, Milan, Mexico City, Teheran or Mumbai and can lead to adverse effects for the population. In such locations, a boundary-layer budget method (Denmead et al., 1996) can be applied during inversion events to estimate the source strength of these substances, since it is then proportional to their concentration increase. We apply this approach to the city of Zurich, Switzerland, which serves as a representative site for Switzerland, which is in turn representative for an industrialized country in Europe with existing mercury regulation and is also party to the Minamata convention. Addi-

tionally, we can profit from our previous studies, where our model has been extensively validated for the city of Zurich and the top-down approach could successfully be applied to quantify emissions of various anthropogenic pollutants. Our previous studies reported top-down derived emissions in Zurich for industrial chemicals, including polychlorinated biphenyls (PCB) (Gasic et al., 2009; Bogdal et al., 2014a; Diefenbacher et al., 2016), flame retardants (Bogdal et al., 2014b; Diefenbacher et al., 2015a), perfluorinated surfctants (Müller et al., 2012; Wang et al., 2012) unintentional combustion byproducts, including poly-

chlorinated dibenzo-p-dioxins and dibenzofurans (Bogdal et al., 2014a), or additives of personal care products such as cyclic methylsiloxanes (Buser et al., 2013). Furthermore, this method is not only applicable to Zurich, but to a multitude of locations and has successfully been applied to various substances as for example PCBs in Chicago, USA, Hazelrigg, UK, Finokalia, Greece, Banja Luka, Bosnia and Hercegovina (MacLeod et al., 2007; Gasic et al., 2010), cyclic methylsiloxanes in Chicaco, USA (Buser et al., 2014), chloro- and hydrofluorocarbon propellants nearby Zurich, Switzerland (Buchmann et al., 2003),

methane in London, UK (Lowry et al., 2001) and St. Petersburg, Russia (Zinchenko et al., 2002). Wherever smog problems



arise such a boundary-layer budget is technically feasible. We hypothesize that mercury has relatively constant emissions and follows the pattern of accumulation during strong inversion periods similarly to the organic pollutants cited before. By developing and applying a box-model for the city of Zurich our aim is to derive the emission source strength of Zurich. Furthermore, we extrapolate our findings to whole Switzerland and compare the calculated emissions to reported emissions from bottom-up

inventories. Finally, we discuss the applicability of our boundary-layer budget approach in a general context.

## 2    Measurements and Methods

### 2.1    Measurements

Gaseous elemental mercury (GEM) concentrations have been measured from December 2013 until December 2015 at the sampling station of the Swiss National Air Pollution Monitoring Network (NABEL), Zurich Kaserne, Switzerland. It is located

in a large courtyard (approximately 9000 m$^2$) in the city center of Zurich (47.38°N, 8.53°E, 409 m above sea level) shielded from highly frequented roads and industrial activities. Since decades, the site has provided continuous monitoring of the major air pollutants and a multitude of meteorological parameters, such as the wind speed, used in this study as a model parameter. Previous work on particulate matter (PM-10) (Hasenfratz et al., 2015; Mueller et al., 2016), nitrogen oxides (NOx) (Mueller et al., 2015) and persistent organic pollutants (Bogdal et al., 2014a; Diefenbacher et al., 2015b, 2016) have shown

that the measurement location provides representative background levels for the city of Zurich and is not affected by acute emissions close to the site. For GEM measurements air was sampled through an inlet and analyzed using a Tekran® 2537X cold vapor mercury analyzer with a detection limit lower than 0.1 ng/m$^3$ stated by the manufacturer. Flow rate was 1.5 l/min and measurements were taken every 5 minutes from alternating cartridges. The instrument was automatically calibrated every 25 hours. Additionally, manual calibrations of the permeation source were performed using an external calibration device

(Tekran® 2505) and comparison measurements were conducted with an instrument identical in construction to ensure data quality. Furthermore, GEM was measured during a single monitoring campaign (January - February 2016) using the same measurement device on a cite in the periphery of Zurich (Zurich Zoo, 47.38°N, 8.58°E, 587 m above sea level), to obtain indications for the background influx of air in Zurich. Methane (CH$_4$) and carbon monoxide (CO) measurements were provided by NABEL (BAFU; EMPA, 2018). CH$_4$ levels are used as a conservative trace gas to compare to GEM levels, while CO is

used as combustion indicator.

### 2.2    Model design

For the model design, we take advantage of the meteorological conditions of a temperature inversion that can occur during high-pressure periods. A phenomenon where – due to the faster cooling of the earth surface – the temperature profile in the atmosphere becomes inverted. Higher in density, cold air resides at the surface and temperature increases with height until the

boundary-layer is reached. This leads to a stratification of air masses, where vertical mixing is very low. In Zurich this phenomenon is enhanced by the valley topography, where cold air drains into the depression. The reduced convective mass transfer



to the warmer air masses above thus restricts the air volume in direct contact with the surface. In summer this phenomenon usually only occurs during the night. The strong soil heating and the resulting thermal lift break up the inversion soon after daybreak. In winter, with lower sun intensity, inversion conditions can prevail for several days up to weeks leading to the well known smog problematic. With steady emissions at the ground level into the smaller volume of the surface layer, an increase

in concentration for air pollutants is observable (Salmond, 2005; MacLeod et al., 2007). This not only accounts for commonly monitored air pollutants such as $CH_4$, CO, $NO_x$, volatil organic compounds (VOCs) and aerosols such as PM-10, but as well for trace chemicals of anthropogenic origin such as persistent organic pollutants (Gasic et al., 2009; Müller et al., 2012; Wang et al., 2012; Bogdal et al., 2014a, b; Diefenbacher et al., 2015a, b), cyclic methylsiloxanes (Buser et al., 2013) or chloro- and hydrofluorocarbons (Buchmann et al., 2003). Under the assumption of constant emissions, the slope of the increase in con-

centrations is proportional to the emission flux. By setting-up a box model, we make use of this circumstance and are able to derive the emission term for the investigated air pollutant. This approach is thereafter also referred to as boundary-layer budget.

## 2.3    Model parametrization

Over the course of the measurement period nine episodes of day-night inversion were identified by visual inspection of the data.

Only events lasting for a minimum of four days were considered. Individual periods show a considerably longer duration of up to 14 days. These events are then reproduced with our previously developed and validated model model. While in our previous studies, the temporal resolution of the air monitoring was significantly limited (resolution of hours to weeks), we profit here from highly resolved GEM data (5 min resolution). We follow the approach to strip the model to the minimum, only processes indispensable to parametrize the conditions at hand are incorporated. This lean model approach prevents over interpretation

of model results and the reduced complexity provides a better conceivability of the model. Thereby, we end up with a model consisting of a single box of air that covers an area $A$, of 10 km × 10 km (100 km$^2$) approximating Zurich's size inhabited by roughly 400'000 people (Fig. 1). The box size was chosen such as to encompass the city's emission sources and has proven to be suitable by previous studies (Wang et al., 2012; Buser et al., 2013; Bogdal et al., 2014b). Based on the national emission inventory (Heldstab et al., 2015) half the mercury emissions are assumed to stem from mixed sources such as stationary com-

bustion, minor industrial activities and houses distributed all over the city. The other halve comes from a waste incineration plant in the middle of the city, where a chimney (height 90 m) leads to a broader distribution. For Zurich this shows an emission profile similar to unintentional combustion byproducts such as polychlorinated dibenzo-p-dioxins and dibenzofurans studied in the work of Bogdal et al. (2014b). Therefore GEM concentrations can be assumed to be homogeneously mixed in the model compartment. The time dependent parameters included in the model are: i) the boundary-layer height (BLH), defining the

volume of the box ii) the wind speed to model advective flux (F) through the box and iii) the background concentrations to quantify the concentrations of the advective flux.

The model is operated dynamically with an hourly resolution. Due to the relatively short periods of maximum 14 days simulated with our model, many parameters usually included in box models can be disregarded. Slow processes such as deposition to soil and water, as well as re-emission from these compartments and atmospheric degradation are excluded due to the short





residence time of mercury within the considered small model region. Our own calculations, with a version including atmospheric degradation reactions, show that losses by degradation are negligible in comparison to the advective fluxes.

The focus of the model is on the emission flux, which is directed to the surface air compartment and is kept constant over the course of an inversion persiod. The goal is to find the emission term that results in a modeled surface GEM concentration

matching best the measured concentrations. The emission flux is the only adjustable parameter in the model, whereas all further model parameters are pre-set and not adapted to improve the fit between model results and field measurements. The residual mean square error (RMSE) is used as a measure for optimization in an iterative fitting process. The emission flux resulting in the lowest RMSE is then applied as the city's source term.

### 2.3.1 Model set-up

The model includes the following three time dependent model parameters.

(i) **The boundary-layer height** (BLH) is used as a measure to define the volumes of the air compartment of the model (Fig. 1). In a first model approach boundary-layer heights are approximated using constant levels for the daylight and nightly period. The height is set to 1500 m for the convective boundary-layer (CBL) during the day from 8 a.m. (UTC+1) until 8 p.m. and lowered to 150 m for the nocturnal boundary-layer (NBL) at 9 p.m.. The NBL level is based on common meteorological con-

ditions (Stull, 1988) and our own experiences of previous work (MacLeod et al., 2007; Gasic et al., 2009; Wang et al., 2012; Buser et al., 2013; Bogdal et al., 2014b). A sharp transition is used between NBL and CBL. These heights determine the box volume of the air layer. Changes in volume of the air compartment are handled such that in case of a decline in BLH, the amount of GEM in the volume difference is transferred out of the box. In case of a rise of the BLH, the GEM concentration in the air compartment is diluted with the corresponding air volume with GEM concentration of background levels.


(ii) **The advection** is determined by the wind speed. Wind speed measurements are conducted above a roof top (33 m.a.g.l.). Work by Benz (1988); Schuhmacher (1992) in Zurich show that measurements at this hight are representative for the mean wind speed for the height profile from 0 - 150 m. As shown later only nightly periods where the BLH is 150 m are relevant to estimate emissions in Zurich. Wind direction is not taken into consideration. Therefore, advection $F$ is always occurring

through a lateral face of the box (Fig. 1) and the flux is obtained by multiplication of its area $A_s$ with the corresponding wind speed $u$, $F = A_s \cdot u$.

(iii) **The background concentration** is set to a steady level of 1.5 ng/m$^3$. It lies in the lower 10% quantile of the whole two year measurement series in Zurich and no adaption was made for nightly backgrounds. In reality background concentrations

are likely to be higher than this level since also at high wind speeds measurements rarely fall below as we show in Fig. S1 and S2. The value of 1.5 ng/m$^3$ lies in the range of what we measured at outskirts of the city of Zurich (Zurich Zoo, median = 1.62, $Q_{0.1}$ = 1.53 ng/m$^3$, $Q_{0.9}$ = 1.77 ng/m$^3$). On the basis of these assumption an upper bound model run regarding the source strength of the city is calculated. To establish a margin, which restricts the source strength with a realistic lower bound,





background concentration are raised to the median concentration of 1.8 ng/m$^3$ for a second emission estimate. We are confident that actual emissions reside within the range of these two model runs.

## 3 Results and discussion

### 3.1 Measurement series

GEM levels measured in Zurich from December 2013 until December 2015, show a median concentration of 1.81 ng/m$^3$ ($Q_{0.1}$ = 1.55 ng/m$^3$, $Q_{0.9}$ = 2.36 ng/m$^3$). The concentration gradient for GEM follows a weak diurnal pattern, similar to $CH_4$ concentrations, however, with a more prominent amplitude. Figure 2 shows the diurnal pattern of GEM, $CH_4$ (i.e. conservative tracer), and CO (i.e. combustion indicator) measurements normalized by their respective mean concentration. The rise in concentrations during night-time and the minimal concentrations during the afternoon are anti-cyclical to the wind speed
and suggest a meteorological cause for the pattern. Stable conditions with lower wind speeds and lower boundary layer height during the night lead to a slight concentrations rise, while higher thermal convection during day-time lower these concentrations. The wind rose plots (Fig. S3) for GEM, $CH_4$ and CO support these findings and our initial hypothesis of a constant source term of GEM for the city. GEM concentrations are thus primarily influenced by the wind speed and the diluting effect of lower background concentrations (Fig. S2). However, one can observe as well slightly lower GEM concentration towards
the weekend with the lowest concentrations on Sundays (Fig. 2). More prominently this is the case for CO, which has sources that are strongly activity related, such as traffic. We therefore deduce that besides the prominent constant sources for GEM, there are as well activity related emissions, but of much smaller scale. The inversion events for the summer extracted from the measurement series show a clear diurnal trend not only for GEM but also for $CH_4$ (Fig. S4), both trace gases stem from constant sources. In winter inversion periods also concentrations for PM-10, CO, $SO_2$, $NO_x$ follow the course of GEM since
combustion related sources have a more constant source term (Fig. S5). These comparisons indicate that the emission flux for GEM and $CH_4$ are constant over time.

### 3.2 Boundary layer budget

To illustrate the model results, we present the period showing longest continuing day-night inversion from 24 June until 06 July 2015 (Fig. 3) as an example. Analogous figures for the other eight inversion periods are shown in the Supplement (Fig. S6-
S14). The figure shows the most important model parameters: the boundary-layer height (Fig. 3a), the wind speed (Fig. 3b), and Fig. 3c the measured GEM concentrations (blue), as well as the model results (red). GEM measurements show a clear diurnal variation with high concentrations of up to more than 3 ng/m$^3$ during the night and lower concentrations during the day. The model results follow this pattern. The emissions strength of the city is observable from the steep concentration increase, when the BLH is lowered. These nocturnal periods with low wind speeds (< 2 m/s) are used to estimate the source term of the city.
During daytime the modeled concentrations are dominated almost entirely by the background concentrations due to the higher BLH and the stronger wind speeds, which create a much bigger flux then the city's GEM emissions.





In general, model results follow the measured concentration suggesting advection and boundary-layer height are indeed enough to describe most of the GEM variation. During day-time model results are slightly lower than measured GEM levels, showing that our model approach covers the most important atmospheric processes occurring on this time scale and successfully reproduce the fate of GEM in the urban air of Zurich. By RMSE reduction we find a GEM emission flux of 4.8 g/hour (Table 1)

for the modeled region in this period, or 42 kg/a when extrapolated to annual emissions for the city of Zurich. Over all nine periods, we find a mean emission of 4.7±0.9 g/hour or 41±8 kg/a (Table 1). The low variance of the emission estimate over all the nine periods from different months and years supports our claim of a constant mercury emission term for the city. Also for three winter periods with long stable inversion conditions (Fig. S15-S17) emission estimates in the same range are observed (4.4±0.8 g/hour, see Table S1).

**3.2.1   Uncertainties of the emission estimates**

The mass balance for mercury in the air compartment of the box ($m$) formulates as follows:

$$\frac{dm}{dt} = V \cdot \frac{dc}{dt} = F_{\text{adv}} \cdot c_{\text{back}} - F_{\text{adv}} \cdot c(t) + E \tag{1}$$

$$\frac{dm}{dt} = u \cdot A_{\text{s}} \cdot c_{\text{back}} - u \cdot A_{\text{s}} \cdot c(t) + E \tag{2}$$

where $V$ [m$^3$] is the Volume of the box, $c_{\text{back}}$ is the background concentration [ng/m$^3$], E is the emission flux [g/s]. The advec-

tive air flux $F_{\text{adv}}$ [m$^3$/s] is calculated from the wind with velocity $u$ [m/s] that flows through the lateral side of the box $A_{\text{s}}$ [m$^2$]. Although the emission estimations presented so far are based on a dynamic time-resolved box model, we introduce here the steady state case for sake of simplification. For the steady state solution ($\frac{dm}{dt} = 0$) the emission flux is $E = u \cdot A_{\text{s}} \cdot (c - c_{\text{back}})$. The problem at hand are of linear nature. Therefore, error handling is straightforward and maximum error bound could be found using linear error propagation (MacLeod et al., 2002) and a given uncertainty in a parameter would at worst results in

an equal uncertainty in the model result. However, the true errors of the parameters are unknown and not all of them would strictly follow a known distribution. The procedure we apply here leads to a more confident error margin.

As mentioned before, the background concentration, $c_{\text{back}}$ of 1.5 ng/m$^3$ used until now is a lower bound for the background concentration that leads to an upper bound estimate for the true GEM emissions. By using 1.8 ng/m$^3$ as background concentration (+20%), which is equal to the median and a high estimate for the background, we are able to set a lower bound

for emissions. These two margins are more helpful in the error characterization than a technical error propagation approach. Following this approach we obtain a lower bound emission of 2.8±1.0 g/hour (Table 1, mean of the lower bound scenarios of the nine periods ± standard deviation) and can thereby narrow down the true GEM emissions for Zurich. According to our findings they must amount to a value between 2.8±1.0 and 4.7±0.9 g/hour. In this range we also see the sensitivity of the background concentration for the model results. As shown a change in $c_{\text{back}}$ by 20% resulted in a mean emission estimate for

all nine periods lower by 40%. The sensitivity ($S = (\Delta O/O)/(\Delta I/I)$, MacLeod et al. (2002)) of $c_{\text{back}}$ ($I$) regarding the mean emission estimate ($O$) amounts to $S = 2$.

After the background concentration, the BLH ist the most sensitive parameter. An increase of the BLH by 10% results in an equally larger emission estimate, $S$ for the BLH equals 1. As mentioned before the height of 150 m for Zurich has been





established in previous model studies from temperature profiles. To test this value and the assumption of a constant height we established an advanced model scenario, where the BLH is derived from a complex numerical weather prediction model COSMO-2, developed by MeteoSchweiz. The BLH is determined both for day and night with an hourly time resolution. The approach for the advanced scenario is presented in detail in the Supplement. All model runs of the advanced scenario are dis-

played in the Figures S6-S14. The emission estimates based on this advanced approach (4.9±1.7 g/hour) are very close to the basic scenario with the fixed BLH presented here (4.7±0.9 g/hour). The basic approach with a fixed BLH of 150 m is therefore justified and to reduce the model complexity we recommend this the basic approach.

Other important model parameters to be discussed here are the model area with a sensitivity of $S = 0.6$ and the wind speed $S = 0.4$. If we again look at a steady state example ($\frac{dm}{dt} = 0$) of equation 2 and rearrange to $c = c_{\mathrm{back}} + E/(u \cdot A_{\mathrm{s}})$, we see that

the ratio between the emission flux $E$ and the advective flux ($u \cdot A_{\mathrm{s}}$) is determining the deviation of the concentration from the background concentration $c - c_{\mathrm{back}}$. During the day, advective fluxes are much larger than the emission term we estimate for Zurich. For the small box in Zurich even for very low wind speeds of 1 m/s, $F_{\mathrm{adv}}$ is 7 times bigger than $E$. Therefore, $c$ is dominated by $c_{\mathrm{back}}$ during the day in our model. The explanatory power of the model is much stronger during the nocturnal inversion periods with low BLH and low wind speeds. During these periods advection has little influence. The model application

domain for the two cases of high and low BLH are displayed in Figure S18. These curves set the domain regarding the GEM concentrations explainable by our model depending on the BLH, the wind speed and the model area. The model area has to be chosen to encompass all important sources and big enough such as to allow for a reasonable time step in the model set up. For large model areas, however, inhomogeneities in the model region could be problematic. Here, we assume homogeneous distribution of GEM in our model region of 10 km by 10 km. As work by Cairns et al. (2011) shows, GEM concentrations in

Toronto do follow a certain distribution and differences in concentrations do occur depending on the measurement location. Our measurement site has been assessed for pollutants with diffuse emissions with passive samplers by comparing various sites throughout the city. The location proved to be a representative study site for an anthropogenic pollutant with diffuse emissions (Diefenbacher et al., 2015b, 2016). As mentioned before mercury emissions in Zurich stem from diffuse sources, which are distributed in the whole city and a waste incineration plant in the city center. Emission estimates derived with our box model

only apply to the whole city or can be averaged by person. Spatially resolved emission estimates are, however, not attainable. Regarding the consistency of the emissions we see from the comparison of the GEM measurements to CO and $CH_4$ levels (see Fig. 2, S4 and S5) in Zurich emissions largely stem from constant sources. Activity related emissions, i.e. from traffic, are a minor contributor. Also from the comparison between summer and winter periods, which are comparable in terms of emission strength, we conclude that the increased combustion activities during the cold winter months are not a large contributor to

overall GEM emissions. Possibly, increased GEM emissions from enhanced combustion activities in winter are compensated by reduced emissions of GEM by evaporation from legacy mercury reservoirs in periods with low ambient temperatures and vice versa in winter.





### 3.3 Implication on emission reporting

Mercury emissions are annually reported by countries signatories of the Protocol on Heavy Metals to the UNECE Convention on Long-Range Transboundary Air Pollution (CLRTP). Swiss national CLRTP inventories for mercury emissions to the atmosphere reported a total of 658 kg/a for the year 2014 (Heldstab et al., 2015). The biggest share, 73% of the emissions, stems

from the energy sector (1A1), of which the majority is allocated to energy industries for public electricity and heat production. Main sources to this energy sector are waste incineration plants. In Switzerland energy recovery from municipal solid waste incineration is mandatory and emissions from waste incineration plants are reported under this category. Other combustion processes mainly in manufacturing industries (1A2:5) make up for 25% of the total emissions. These numbers and categorization into individual sectors and subcategories as shown in Table 2 and set the basis for the allocation to the global emission grid of

EMEP for Switzerland. The grid shows spatial resolved emissions with a $0.1° \times 0.1°$ (approx. 10 km × 10 km) resolution. The allocation rules for the emissions rely mostly on population density and vary from one to another source category. Depending on the source category, different weightings on the prevailing employment sectors are installed. For Zurich, the gridded emission report an emission flux of 18 kg for the year 2014 (18 kg for 2015). Our boundary-layer budget approach results in a GEM emission flux between 24 and 41 kg for Zurich. These findings suggest emission of about double the amount allocated to Zurich

by the rule set for the EMEP report. If we would apply the same allocation factors to the model results we would come up with national GEM emission of 934 to 1581 kg/a, i.e. clearly higher than reported by the authorities. When we use population data only as a criterion, a scaling factor of 20.5 would be appropriate, considering a population of approximately 400'000 residence in the modeled area and a Swiss population of 8.2 million people. This approach would amount to emission of 494 to 837 kg GEM per year. In comparison to the 658 kg/a of the Swiss CLRTP report, these results lie in a very acceptable range and show

that the approach explained here can be used to validate national reporting. Moreover, from our results we suggest that the allocation formula for mercury for the EMEP grid should be adjusted such that population data is given more weight over other parameters. For Zurich we find a per capita emission of 0.06 to 0.10 g/a per person. This estimate is somewhat lower than the European per capita emission of 0.19 g/a reported in the AMAP/UNEP (2013) background report.

### 4    General applicability and conclusions

The boundary-layer approach presented here is based on atmospheric inversion. This phenomenon, however, is not unique only to Zurich, but can be applied to a wide range of cities and industrial complexes of different size all over the world. Diurnal variability with higher nighttime GEM concentrations has for example been observed in Southern England (Lee et al., 1998); Seoul, Korea (Kim and Kim, 2001); Guiyang, China (Feng et al., 2004), and Beijing and Guangzhou, China (Wang et al., 2007). By adapting the model to the localities, emission estimates are feasible and can support authorities in the set-up and

improvement of emission inventories. Indicators for inversion are manifold and are manifest by temperature inversion, increases in pollution levels or morning fog. The adaptiveness of the box model approach is displayed in Fig. 4. It shows a graphical representation of the steady-state equation for the emissions ($E = u \cdot A_s \cdot (c - c_{back})$). We argue that steady-state is reached to a reasonable degree when atmospheric conditions are stable for a period of several hours, depending though on the size of the





box and the wind speed. The nomogram (Fig. 4) can be used in order to quickly estimate the emission strength of a city under the assumption of steady-state conditions. Two exemplary cities, Beijing, China and Denver, CO, USA of different size are shown in the graph together with Zurich. All of them are located in a valley, a beneficial characteristic but not a requirement to the model. The starting point to read the graph marks the wind speed measured during an inversion period on the bottom

right side of the graph. The graph to the left than gives the proportionality between the difference in GEM concentrations of the measurements and the background $(c - c_0)$ on the abscissa and the GEM emission strength on the axis of ordinate. A walk through example is given for Beijing with a wind speed of 1.5 m/s. Continuing in a straight path upwards (see gray vertical line on the right part of Fig. 4) the line given by the size of Beijing is reached (black dotted line). The box length for a city and the BLH define the lateral side of the box $A_s$. Here, we use our standard BLH of 150 m. From this point one draws a

horizontal line to the left until the left plot is reached. A GEM concentration difference of 2.6 ng/m$^3$ results in an emission estimate of 77.5 g/h. Following this example emissions strength of other cities can quickly be estimated according to size of their respective box model, the measured wind speeds and GEM concentrations. The colored lines show the situation in Zurich, i.e. the emission estimations with a wind speed of 0.5 m/s (red lines), 1 m/s (green lines), 2 m/s (blue lines), and 4 m/s (purple line). The distribution of the daily minimum wind speeds and the daily maximum concentration differences measured in Zurich

during inversion are given by the density curves at the bottom. These curves set the bandwidths for GEM emissions predicted by the model. The emission estimates of the high and low backgrounds are given by the gray straight lines 2.8 and 4.7 g/hour. The comparison thereof with the emissions from the steady-state box model presented here shows a reasonable accordance.

An other entry point to read the graph is the emissions. For Beijing emission estimates amount to 775 g/h (6.79 t/a) (Streets et al., 2005; Wu et al., 2006; Zhou et al., 2010). Since the formula is all linear both axis of the left graph can as well be multiplied

by 10, so the entry is congruent to 77.5 g/h. Concentration difference for a wind speed of 1.5 m/s than amounts to 26 ng/m$^3$ (2.6 ng/m$^3 \times$ 10), which is a reasonable value for average concentrations in Beijing, considering the GEM measurements reported for the city (4 - 54 ng/m$^3$) (Wang et al., 2007; Zhou et al., 2010). Based on this preliminary assessment a more extensive box model study could be conducted, taking into account specific measurements and as well technical aspects such as the the bigger air compartment, where the inhomogeneities have to been addressed with multiple measurement locations.

We believe the boundary-layer budget approach presented here is a valuable contribution to the demand for mercury emission inventories by the UN Minamata Convention on Mercury (Article 19, 1.a). The low computational requirements a box model poses and its broad applicability make it a readily available tool that is needed in narrowing down the broader scope of common bottom-up emission estimates. The use of passive samplers for mercury, which allow a cost effective and broad spacial coverage in ambient air monitoring, in combination with box models such as ours pose a great opportunity not only for

model refinement, but also for the applicability to other domains. The fields of application, however, are not limited to mercury alone, other compounds are as well suited for emission estimates by a box model. Furthermore, besides emissions, sinks – a hot topic in mercury research – can also be quantified with the presented boundary-layer budget method.





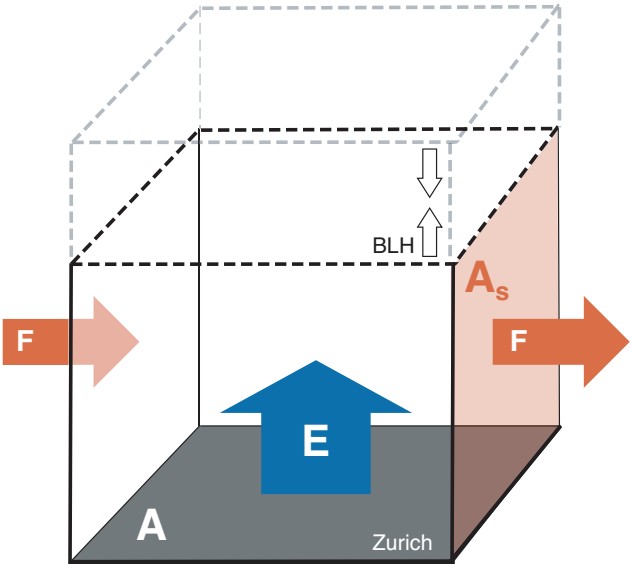

**Figure 1.** Box model used to to estimate GEM emissions $E$ of Zurich. $A$ represents the area of the base and $A_s$ the area of the lateral side of the box. The variable boundary-layer height (BLH) determines the height of the box and $F$ the advective flow.

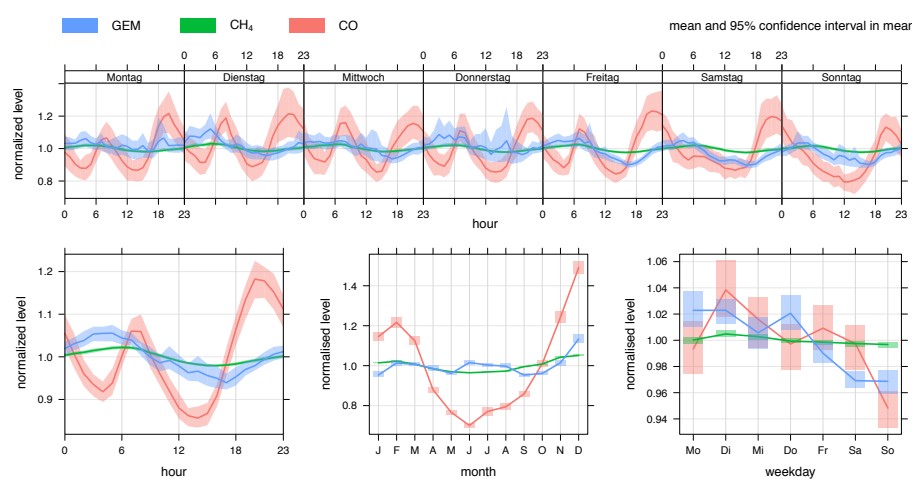

**Figure 2.** Comparison between normalized GEM, $CH_4$ and $CO$ levels (divided by overall mean) showing the temporal variation over a weekly, daily, monthly and weekday course. Mean values and 95% confidence intervals are shown. [Source: NABEL (FOEN and Empa) BAFU; EMPA (2018)]





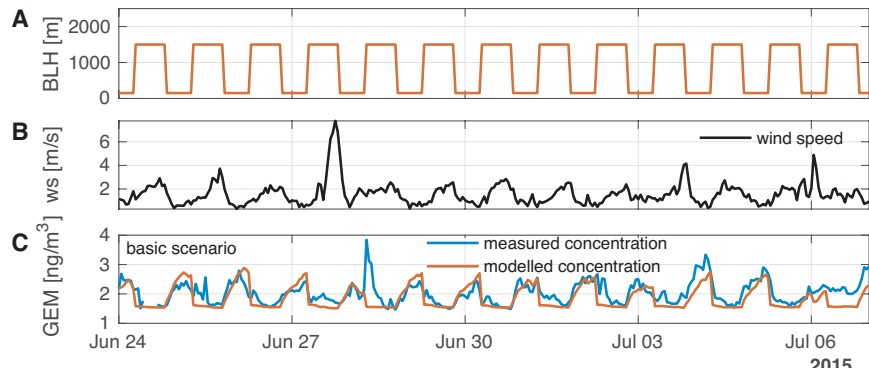

**Figure 3.** Exemplary period from June 24 until July 06 2015 that shows a day-night pattern for ground inversion. **A** shows the boundary layer heights for both the basic scenario (red). **B** shows the wind speed (ws). **C** shows GEM measurements in blue and the basic model results in red.

**Table 1.** The emission estimates of GEM in Zurich, Switzerland are shown for all nine observed summer periods.

| start | days | emissions [g/hour] | |
|---|---|---|---|
| | | upper bound | lower bound |
| 06/06/2014 | 4 | 5.7 | 3.8 |
| 16/07/2014 | 5 | 4.4 | 2.5 |
| 05/09/2014 | 4 | 4.0 | 2.4 |
| 05/03/2015 | 6 | 3.5 | 1.7 |
| 18/05/2015 | 5 | 3.4 | 1.1 |
| 24/06/2015 | 14 | 4.8 | 3.0 |
| 11/07/2015 | 13 | 5.9 | 38 |
| 02/08/2015 | 5 | 5.7 | 3.7 |
| 28/08/2015 | 4 | 4.5 | 2.7 |
| mean | | 4.7±0.9 | 2.8±1 |
| annual [kg/a] | | 41±8 | 24±8 |





**Table 2.** Swiss national inventory for mercury emissions in 2014 as submitted under the UNECE Convention on Lang-range Transboundary Air Pollution. (NFR: nomenclature for reporting emission categories, IPPU: industrial processes and product use)

| Hg emissions 2014 | | national | this study |
|---|---|---|---|
| NFR | category | [kg/a] | [kg/a] |
| 1 | power | 482 | |
| 1A | fuel combustion | 482 | |
| A1 | energy indust. | 279 | |
| A2:5 | misc. | 162 | |
| 2 | IPPU | 78 | |
| 5 | waste | 35 | |
| 6 | other | 63 | |
| | total | 658 | 574 - 951 |

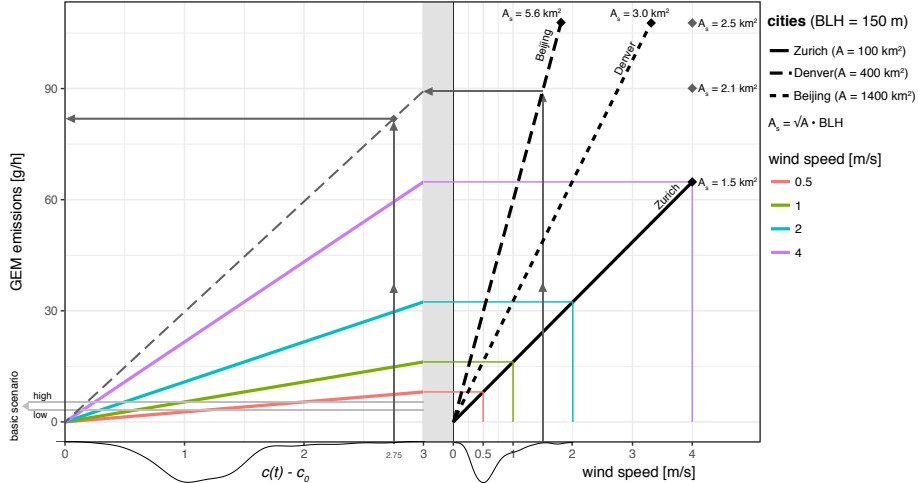

**Figure 4.** Graphical representation of the stead-state formula $E = u \cdot A_s \cdot (c - c_{back})$ to estimate emissions in a box model. In color, starting at different windspeed it shows the corresponding GEM emission estimate for Zurich in relation to the concentration difference between measurement $c$ and background $c_0$. The results of the basic emission scenario are given as gray lines. The gray shading highlights the range in concentration differences obtained in Zurich. An example how emission are estimated for an other city is given for Beijing, with a wind speed of 1.5 m/s and 2.6 ng/m$^3$ concentration difference.



*Author contributions.* Basil Denzler and Christian Bogdal planed the study and executed the measurements and prepared the manuscript. Cyrill Kern, Anna Tobler and Jing Huo worked on the model development during their masters course. Konrad Hungerbühler supervised the project.

*Competing interests.* The authors declare that they have no conflict of interest.

5   *Acknowledgements.* We would like to thank Stephan Henne (EMPA, Dübendorf) for the helpful advice he provided. Furthermore, we acknowledge the Swiss National Air Pollution Monitoring Network (NABEL) and the Federal Office for Meteorology and Climatology (MeteoSwiss) for providing measurement and meteorological data. We thank the Swiss Federal Office for the Environment (FOEN) for the project funding (grant numbers 00.0248.P2/M371-4632, 14.0039.KP/N412-1043).



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
