# Peer review of "Boundary-layer investigation"

_Atmospheric Chemistry and Physics, 2018_

## Referee Comment (RC1) · Anonymous Referee #1 · 2 Aug 2018

Denzler and coworkers present a top-down approach to estimate urban mercury emissions from ground-based measurements. Their approach is a nice and simple boundary-layer mass balance method applied during periods of temperature inversion and low wind speeds when the measured urban concentrations are most sensitive to local emissions. Using two years of measurements, they calculate gaseous elemental mercury emissions in Zurich and compare it to those reported in the Swiss national emissions inventory. The authors also provide a convenient tool based on their approach to calculate gaseous elemental mercury emissions in other cities.

It is important to quantify the anthropogenic emissions of mercury to understand its effects on the biogeochemical cycling of mercury and to build the knowledge needed for the success of regional and global efforts to lower the human health burden of mercury.

[Figure]

Bottom-up emissions inventories are uncertain and need to be checked against top-down estimates, as has been done in this study. The top-down method described here will certainly be of much interest to readers of ACP who study mercury and also to those who are working to quantify emissions of similar pollutants. The study is scientifically sound, well written, and presents the relevant data supporting their conclusions.

I have divided my further comments into two categories as follows: Major comments: (i) The authors seem to have completely overlooked gas and particle-bound oxidized mercury. There is ample evidence that a significant fraction of mercury emissions are in these forms (e.g. Zhang et al., 2016). It is important that the authors discuss the effect of neglecting oxidized mercury on their emissions estimate and its comparison with estimates of (total) mercury emissions.

(ii) In section 4, the authors provide a tool in the form of a nomogram that can supposedly be adapted by nonspecialists to calculate emissions elsewhere. This is indeed useful, but I am concerned that the conditions under which this tool is broadly applicable (or not applicable) are not clearly laid out. I recommend that a more objective description of this be presented in this section. It could include for example a threshold for the meteorological parameters for which their method works, the general characteristics of the site that measures the urban background, what if the stacks of point sources are taller than 150 m, etc.

Minor comments: Figure 2 shows the weekly and monthly variation of mercury concentrations and that of other species. The authors use this figure to demonstrate that mercury emissions are constant in time. However, the relatively high background concentration of mercury makes the variations in local concentrations seem small. It would be more insightful to subtract the background and then show how local mercury concentrations vary in time. It is not clear why the deposition, emissions from land and water, and oxidation of mercury can be neglected in the model. This needs to be better discussed with relevant citations. In Section 2.1, a second measurement site on the outskirts of Zurich is mentioned. But those measurements are not discussed in the paper. I think they could provide valuable constraints on the spatial contrasts in mercury and help support their assumption that deposition and chemistry can be neglected. Page 4, line 14: "...were identified by visual inspection of the data." Which data? Page 3, line 30: "boundary-layer *top* is reached" There are several spelling errors that I assume will be corrected in the production process if the paper is accepted.

References: Zhang, L., Wang, S., Wu, Q., Wang, F., Lin, C.-J., Zhang, L., Hui, M., Yang, M., Su, H., and Hao, J.: Mercury transformation and speciation in flue gases from anthropogenic emission sources: a critical review, Atmos. Chem. Phys., 16, 2417-2433, https://doi.org/10.5194/acp-16-2417-2016, 2016.

---

## Short Comment (SC1) · 28 Sep 2018

The manuscript aims to develop and validate an emission quantification approach, with application to gaseous elemental mercury (GEM) in Zurich. My first impression was that the method developed is simply the well-known one box / urban column model (Daniel Jacob, 1999: Chapter 3), with addition of fixed day and night boundary layer heights, and that similar but more detailed treatment is given, for example, by Jin and Demerjian 1993 (Atmos. Environ.). Closer reading shows there is several differences, with use during inversion conditions, a simple boundary layer height (BLH) parameterization scheme, and a graphical tool. To this reader, the most interesting aspect was the BLH discussion. The proposed scheme appears the same as previous work by the authors (Bogdal et al. 2014b) but is enhanced by comparison to a more advanced model. Still, this was mostly relegated to the supplement, and the validation against the advanced model is not so convincing since that model does show much better agreement for GEM concentrations, despite the better BLH representation. The paper should be clearer about what is novel since box models are well tread ground, and perhaps reorganized to expand on those aspects. The introduction could also do a better job setting up how the work fits in with past literature boundary layer budgets and urban emissions quantification.

One specific comment: what is the height above ground for the NABEL measurement? A homogeneously mixed assumption is mentioned on Page 4, but is this really justified, or simply is necessary since there is only one measurement location available? In the vertical, the stratified atmospheric inversion conditions could lead to error/bias when using a (presumably near surface) concentration monitor; the incinerator chimney source which is at 90 m also may not be represented by the measured GEM and lead to errors in the emissions estimate.

---

## Referee Comment (RC2) · Anonymous Referee #2 · 2 Oct 2018

General comments:

The topic of the manuscript is relevant to this journal. The discussion and conclusions are based on the results obtained. There are some points to be revised before the acceptance of the manuscript. They are:

1) Discussion on Hg(II)

2) Quantitative measures to evaluate the model fit

3) Typos

Specific comments:

**1 Discussion on Hg(II)**

[Figure]

3.3 Implications on emission reporting, Table 2

Please clarify whether the Swiss national CLRTP inventories for mercury emissions include only Hg(0) emissions or they include both Hg(0) and Hg(II) emissions. In the latter case, omission of Hg(II) in the boundary layer budget model needs to be discussed.

**2 Quantitative measures to evaluate the model fit**

Figure 3 C, Figure S6-S17

As to the comparison between the measured concentrations and the modelled concentrations, visual comparison of the graphs are used in figure 3C and figure S6-S17. However, it is not so easy to evaluate the goodness of the fit in a quantitative manner thorough the visual inspection.

The reviewer recommends the authors to present some quantitative indicators that can be used to judge the goodness of the fit. For example, the RMSE used for the optimization might be presented along with the RMSE calculated for models with fixed BLH at 1500m (no inversion). The RMSE might be also useful for comparing basic scenarios and advanced scenarios.

**3 Typos**

**3-1: Table 1 and Table S1**

The lower bound emission for 11/07/2015 is "38" g/hour, which seems too large and is not consistent with the mean lower bound emission of 2.8 g/hour.

**3-2: Page 4, line 16**

"and validated model model" -> "and validated model"

**3-3: Page 7, line 32**

"the BLH ist the most sensitive" -> "the BLH is the most sensitive"

---

## Author Comment (AC2) · 27 Nov 2018

**Referee comment in bold**, reply in plain text.

A revised and highlighted version of the manuscript is available in the supplementary material.

**The topic of the manuscript is relevant to this journal. The discussion and conclusions are based on the results obtained. There are some points to be revised before the acceptance of the manuscript.**

The referee has recognized the relevance of this work for this journal and we readily address the issues raised to meet the referee's standards.

[Figure]

**#1 Discussion on Hg(II) 3.3 Implications on emission reporting, Table 2 Please clarify whether the Swiss national CLRTP inventories for mercury emissions include only Hg(0) emissions or they include both Hg(0) and Hg(II) emissions. In the latter case, omission of Hg(II) in the boundary layer budget model needs to be discussed.**

The Swiss national CLRTP inventories do encompass mercury and mercury compounds and as such also oxidized mercury species. The discussion and the introduction of the manuscript has, therefore, been revised accordingly. (page 9, line 25)

**#2 Quantitative measures to evaluate the model fit Figure 3 C, Figure S6-S17 As to the comparison between the measured concentrations and the modelled concentrations, visual comparison of the graphs are used in figure 3C and figure S6-S17. However, it is not so easy to evaluate the goodness of the fit in a quantitative manner thorough the visual inspection. The reviewer recommends the authors to present some quantitative indicators that can be used to judge the goodness of the fit. For example, the RMSE used for the optimization might be presented along with the RMSE calculated for models with fixed BLH at 1500m (no inversion). The RMSE might be also useful for comparing basic scenarios and advanced scenarios.**

The RMSE for the model fit have been added to table S2 in the supplementary material. We see this adjustment is a valuable contribution that further substantiates the decision made to focus on the basic scenario.

**#3 Typos**

**#3-1: Table 1 and Table S1 The lower bound emission for 11/07/2015 is "38" g/hour, which seems too large and is not consistent with the mean lower bound emission of 2.8 g/hour.**

The lower bound emission for 11/07/2015 have been corrected to the value of 3.8 g/hour.

**#3-2: Page 4, line 16 "and validated model model" -> "and validated model"**

Corrected

**#3-3: Page 7, line 32 "the BLH ist the most sensitive" -> "the BLH is the most sensitive"**

Corrected

Please also note the supplement to this comment:
https://www.atmos-chem-phys-discuss.net/acp-2018-402/acp-2018-402-AC2-supplement.pdf

**Supplement:**

[revised manuscript text omitted]

---

## Author Response (AR1)

**Reply to Anonymous Referee #1**

**Referee comment in bold**, reply in plain text, *modified text for manuscript in italics*.

**Denzler and coworkers present a top-down approach to estimate urban mercury emissions from ground-based measurements. Their approach is a nice and simple boundary-layer mass balance method applied during periods of temperature inversion and low wind speeds when the measured urban concentrations are most sensitive to local emissions. Using two years of measurements, they calculate gaseous elemental mercury emissions in Zurich and compare it to those reported in the Swiss national emissions inventory. The authors also provide a convenient tool based on their approach to calculate gaseous elemental mercury emissions in other cities.**

**It is important to quantify the anthropogenic emissions of mercury to understand its effects on the biogeochemical cycling of mercury and to build the knowledge needed for the success of regional and global efforts to lower the human health burden of mercury. Bottom-up emissions inventories are uncertain and need to be checked against top-down estimates, as has been done in this study. The top-down method described here will certainly be of much interest to readers of ACP who study mercury and also to those who are working to quantify emissions of similar pollutants. The study is scientifically sound, well written, and presents the relevant data supporting their conclusions.**

We would like to thank the referee for the positive review and for recognizing the importance of the field of atmospheric mercury research and the need to further constrain bottom-up mercury inventories.

**(i) The authors seem to have completely overlooked gas and particle-bound oxidized mercury. There is ample evidence that a significant fraction of mercury emissions are in these forms (e.g. Zhang et al., 2016). It is important that the authors discuss the effect of neglecting oxidized mercury on their emissions estimate and its comparison with estimates of (total) mercury emissions.**

Referee #1 raises an important aspect of atmospheric mercury, which we so far have not discussed in the manuscript mainly due to the lack of data on swiss mercury emissions in this regard. However, we see the need to inform the reader about our thoughts and assumption on oxidized mercury. We have thus introduced several parts addressing the topic of mercury speciation.

First of all, the composition of atmospheric mercury has been addressed in the introduction (page 1, line 22). Reference [1] added.

Furthermore, a comment on the major point source in the model area has been made in the section of model parametrization. (page 5, line 2) References [2, 3] have been added to support the claims in this section.

**(ii) In section 4, the authors provide a tool in the form of a nomogram that can supposedly be adapted by nonspecialists to calculate emissions elsewhere. This is indeed useful, but I am concerned that the conditions under which this tool is broadly applicable (or not applicable) are not clearly laid out. I recommend that a more objective description of this be presented in this section. It could include for example a threshold for the meteorological parameters for which their method works, the general characteristics of the site that measures the urban background, what if the stacks of point sources are taller than 150 m, etc.**

A section, discussing the limitation of the boundary-layer budget approach has been added, as has been recommended by the Referee #1. (page 11, line 17)

**Minor comments:**

**Figure 2 shows the weekly and monthly variation of mercury concentrations and that of other species. The authors use this figure to demonstrate that mercury emissions are constant in time. However, the relatively high background concentration of mercury makes the variations in local concentrations seem small. It would be more insightful to subtract the background and then show how local mercury concentrations vary in time.**

Momentarily, in Figure 2 we show a relative concentration for all the trace gases CO, $CH_4$ and GEM. For a comparison of the three gases relative concentrations are necessary. A background subtraction does not change the variation. It only changes the scale of the y-axis. However, the proportions between CO, $CH_4$ and GEM stay the same. We therefore argue to maintain the current representation, which has the advantage of clarity and best interoperability. Furthermore, as such we do not introduce any assumptions made regarding the GEM background concentrations into the graph.

**It is not clear why the deposition, emissions from land and water, and oxidation of mercury can be neglected in the model. This needs to be better discussed with relevant citations.**

Deposition, emissions from land and water, and oxidation of GEM are without a doubt important processes for the description of the atmospheric fate of mercury. The only reason we can neglect these processes is that they are relatively slow. Considering our small model area, the residence time within this box is short (1h for windspeed of 3 m/s) When comparing the fluxes produces by these processes within the small model area they are negligible in comparison to the strong advective flux. The description on this has been specified. (page 5, line 12)

**In Section 2.1, a second measurement site on the outskirts of Zurich is mentioned. But those measurements are not discussed in the paper. I think they could provide valuable constraints on the spatial contrasts in mercury and help support their assumption that deposition and chemistry can be neglected.**

A paragraph in the measurement results section has been added to the manuscript discussing the results from the second measurement location as suggested by the referee. (page 7, line 4)

**Page 4, line 14: ". . .were identified by visual inspection of the data." Which data?**

The sentence has been changed to: *Over the course of the measurement period nine episodes of day-night inversion were identified by visual inspection for the criteria of strong day/night inversion.* (page 4, line 17)

**Page 3, line 30: "boundary-layer \*top\* is reached"**

The line has been changed according to the suggestion.

**Additional References**

[1]     Gay, D. A., Schmeltz, D., Prestbo, E., Olson, M., Sharac, T., and Tordon, R.: The Atmospheric Mercury Network: measurement and initial examination of an ongoing atmospheric mercury record across North America, Atmospheric Chemistry and Physics, 13, 11 339–11 349, https://doi.org/10.5194/acp-13-11339-2013, http://www.atmos-chem-phys.net/13/11339/2013/, 2013.

[2]     van Velzen, D., Langenkamp, H., and Herb, G.: Review: Mercury in waste incineration, Waste Management & Research, 20, 556–568, https://doi.org/10.1177/0734242X0202000610, 2002.

[3]     Zhang, L., Wang, S., Wu, Q., Wang, F., Lin, C. J., Zhang, L., Hui, M., Yang, M., Su, H., and Hao, J.: Mercury transformation and speciation in flue gases from anthropogenic emission sources: A critical review, Atmospheric Chemistry and Physics, 16, 2417–2433, https://doi.org/10.5194/acp-16-2417-2016, www.atmos-chem-phys.net/16/2417/2016/, 2016.

**Reply to Anonymous Referee #2**

**Referee comment in bold**, reply in plain text.

**The topic of the manuscript is relevant to this journal. The discussion and conclusions are based on the results obtained. There are some points to be revised before the acceptance of the manuscript.**

The referee has recognized the relevance of this work for this journal and we happily address the issues raised to meet the referee's standards.

**#1 Discussion on Hg(II)**

**3.3 Implications on emission reporting, Table 2**

**Please clarify whether the Swiss national CLRTP inventories for mercury emissions include only Hg(0) emissions or they include both Hg(0) and Hg(II) emissions. In the latter case, omission of Hg(II) in the boundary layer budget model needs to be discussed.**

The Swiss national CLRTP inventories do encompass mercury and mercury compounds and as such also oxidized mercury species. The discussion and the introduction of the manuscript has, therefore, been revised accordingly. (page 9, line 25)

**#2 Quantitative measures to evaluate the model fit**

**Figure 3 C, Figure S6-S17**

**As to the comparison between the measured concentrations and the modelled concentrations, visual comparison of the graphs are used in figure 3C and figure S6-S17. However, it is not so easy to evaluate the goodness of the fit in a quantitative manner thorough the visual inspection.**

**The reviewer recommends the authors to present some quantitative indicators that can be used to judge the goodness of the fit. For example, the RMSE used for the optimization might be presented along with the RMSE calculated for models with fixed BLH at 1500m (no inversion). The RMSE might be also useful for comparing basic scenarios and advanced scenarios.**

The RMSE for the model fit have been added to table S2 in the supplementary material.

**#3 Typos**
**#3-1: Table 1 and Table S1**

**The lower bound emission for 11/07/2015 is "38" g/hour, which seems too large and is not consistent with the mean lower bound emission of 2.8 g/hour.**

The lower bound emission for 11/07/2015 have been corrected to the value of 3.8 g/hour.

**#3-2: Page 4, line 16**
**"and validated model model" -> "and validated model"**

corrected

**#3-3: Page 7, line 32**
**"the BLH ist the most sensitive" -> "the BLH is the most sensitive"**

Corrected

**Reply to Levi Golston**

**Referee comment in bold**, reply in plain text, *modified text for manuscript in italics*.

**One specific comment: what is the height above ground for the NABEL measurement?**

The height above ground is 2 m. A remark has been added to the manuscript (page 3, line 19).

**A homogeneously mixed assumption is mentioned on Page 4, but is this really justified, or simply is necessary since there is only one measurement location available? In the vertical, the stratified atmospheric inversion conditions could lead to error/bias when using a (presumably near surface) concentration monitor; the incinerator chimney source which is at 90 m also may not be represented by the measured GEM and lead to errors in the emissions estimate.**

We acknowledge this remark. A homogeneously mixed air compartment is in fact a prerequisite to our model since we have only one measurement just above ground below the boundary layer. Golston correctly pointed this out and we have specified this point in the model description (page 4, line 32). A possible stratification of the lower air masses, therefore, is a source of uncertainty to our model study, but not necessarily a source for error.

[revised manuscript text omitted]